# Access to perinatal healthcare in minority Anglophones: Hospital type and birth outcomes

**Nathalie Auger** [1,2]*, **Marianne Bilodeau-Bertrand**[3], **Nahantara Lafleur**[3]

**1** Department of Social and Preventive Medicine, School of Public Health, University of Montreal, Montreal, Quebec, Canada, **2** Department of Epidemiology, Biostatistics and Occupational Health, McGill University, Montreal, Quebec, Canada, **3** University of Montreal Hospital Research Centre, Montreal, Quebec, Canada

* nathalie.auger@inspq.qc.ca

## Abstract

### Objectives

We assessed the relationship between hospital characteristics and risk of adverse birth outcomes among minority Anglophones in Montreal, Canada.

### Methods

The study included 124,670 births among Anglophones in metropolitan Montreal between 1998 and 2019. We estimated risk ratios (RR) and 95% confidence intervals (CI) for the association between hospital characteristics, including residential proximity to hospitals and language in which medical services are provided, and risks of preterm birth and stillbirth. Models were adjusted for maternal socioeconomic status and other characteristics.

### Results

In this study, 8% of Anglophones had a preterm birth and 0.4% a stillbirth. Anglophone women who delivered at a farther French hospital had a greater risk of stillbirth (RR 1.67, 95% CI 1.28–2.18) than preterm birth (RR 1.21, 95% CI 1.14–1.30), compared with delivery at hospitals closer to home. In contrast, delivery at a farther English hospital was associated with similar risks of stillbirth (RR 1.36, 95% CI 1.08–1.71) and preterm birth (RR 1.36, 95% CI 1.29–1.44). The greater risk of stillbirth with delivery at a farther French hospital, versus greater risk of preterm birth at a farther English hospital, remained present in analyses stratified by maternal age, education, material deprivation, and region of origin.

### Conclusion

Minority Anglophones in Montreal who travel to a farther French hospital for delivery have a greater risk of stillbirth than Anglophones who travel to a farther English hospital. This novel observation suggests the need to determine if access to perinatal healthcare in a woman's language may help reduce the risk of stillbirth.

**Data Availability Statement:** The data underlying the results presented in the study are available from the Institut de la statistique du Québec (https://statistique.quebec.ca/research/#/accueil).

**Funding:** Author N.A. was funded by Health Canada via the McGill Training and Retention of Health Professionals Project (M3.21.23.AUG. INSPQ.01, https://www.mcgill.ca/dialoguemcgill/ trhpp/m2intro) and Fonds de recherche du Québec-Santé (296785, https://frq.gouv.qc.ca/en/ health/). The funders had no role in study design, data collection and analysis, decision to publish, or preparation of the manuscript.

**Competing interests:** The authors have declared that no competing interests exist.

## Introduction

Suboptimal access to perinatal care is associated with preterm birth and stillbirth among ethnocultural minorities with language barriers [1–4]. Communication may be one of the most important challenges for minorities to overcome, as perinatal healthcare services frequently require an understanding of complicated medical terminology [5]. In a systematic review, migrant women highlighted the difficulty finding healthcare services, obtaining pregnancy-related information, and understanding procedures that required patient consent [5]. Minority women also may not recognize the importance of disclosing potentially serious symptoms to care providers [5]. Optimal healthcare is particularly necessary at the time of delivery to prevent preterm birth and stillbirth. These adverse birth outcomes are sensitive to healthcare and are potentially preventable through optimal management [6, 7].

To improve communication with providers, women from cultural minorities may choose to travel farther to obtain perinatal care in their language or have the benefit of translation services. This is particularly the case in Montreal, Quebec, where Anglophones are a minority and represent only 13.2% of the population [8]. Most hospitals in Montreal are Francophone, catering to the French majority. Nevertheless, some hospitals are fully Anglophone and provide services primarily in English [9]. To access these hospitals, however, Anglophones may have to travel farther from home. Studies have shown that longer travel times between the residence and birth hospital may be associated with preterm birth or perinatal mortality [10–12]. In this study, we investigated whether Anglophone women who traveled farther for English perinatal healthcare had a greater risk of preterm birth or stillbirth than women who traveled to a farther French hospital or delivered closer to home.

## Materials and methods

### Data

We analyzed a population-based cohort of 124,670 births among Anglophone women residing in the metropolitan centre of Montreal, Quebec, Canada between 1998 and 2019. Anglophones are the most prevalent linguistic minority in Quebec after Francophones [8]. We used data from live birth and stillbirth registration certificates that cover the entire population, and contain information on maternal mother tongue, place of residence, hospital of birth, and gestational age at delivery. We restricted the analysis to mothers who reported English as their mother tongue and resided in metropolitan Montreal. We excluded deliveries that occurred at home or in birthing centres because our primary interest was the characteristic of the hospital. Over 98% of births take place in hospital in Quebec. Healthcare is publicly funded and there is no cost to access perinatal services.

### Exposure

The main exposure measure was the primary language of the hospital used for delivery. Women could deliver at the hospital closest to their residence or at a farther hospital that offered either English or French perinatal services. Although French hospitals can provide care in English, staff may not be as fluent or find it as simple to identify urgent care needs. Ease of communication may not be as fluid for Anglophone women requiring tertiary care at a French hospital.

We obtained information on the primary language of hospitals from the Quebec Ministry of Health and Social Services [9]. To identify the closest hospital, we used ArcGIS version 10.7.1 (Esri Inc., Redlands, CA) to geocode the residential postal code and the postal codes of hospitals. Postal codes in Montreal cover small areas or one side of a street block. We then

used road maps to calculate the street distance between the centre of each residential postal code and each hospital postal code [13]. The hospital with the shortest distance from the residence was considered the closest hospital.

We grouped women under one of three categories: women who delivered at a farther English hospital; women who delivered at a farther French hospital; and women who delivered at the hospital closest to their residence. Women may have different reasons for travelling farther, including need for higher level care or personal preference. However, women who travel to a farther English hospital are more likely to prefer perinatal care in their mother tongue. Anglophone women who travel to a farther French hospital may have less choice or no language preference. We compared both these groups with women who delivered at the hospital nearest to their residence, the reference in this study.

## Birth outcomes

The two main study outcomes were preterm birth and stillbirth. Preterm birth and stillbirth are indicators of perinatal health that are sensitive to social determinants and healthcare services [6, 7]. Both are adverse outcomes, although some stillbirths may be prevented by having a lower threshold for preterm delivery [6]. Stillbirths include fetal deaths that occur antepartum or intrapartum. Antepartum fetal deaths can potentially be prevented with labor induction, whereas intrapartum deaths can be prevented with cesarean delivery.

We defined preterm birth as delivery before 37 completed weeks of gestation, and stillbirth as the intrauterine death of a fetus weighing 500g or more that was not due to voluntary termination. There were 15 births with missing gestational age, which we excluded from the analysis of preterm birth.

## Covariates

We included covariates that potentially influenced the association between hospital characteristics and risk of adverse birth outcomes. These included maternal age ($<$25, 25–34, $\geq$35 years), parity (0, 1, $\geq$2), legal civil status (married, single or common-law, unknown), maternal region of origin (Canadian born, foreign born), education (no high school diploma, high school diploma with or without vocational training, university, unknown), and material deprivation (high, middle-high, middle, middle-low, low, unknown). The level of material deprivation was derived from census data and expressed as composite index of the employment rate, proportion of individuals without a high school diploma, and average personal income of neighborhoods [14].

## Data analysis

We examined descriptive characteristics of births (n, %). We used log-binomial regression to estimate risk ratios (RR) and 95% confidence intervals (CI) for the association between hospital type and the risk of preterm birth and stillbirth. Models were adjusted for maternal age, parity, civil status, maternal region of origin, education, and material deprivation.

We investigated if maternal characteristics, including age, region of origin, education, and deprivation, modified the association between birth hospital and risk of adverse outcomes. To do so, we first determined if maternal age, origin, education, and deprivation were associated with the type of hospital used for delivery. We subsequently assessed if these maternal characteristics were associated with the risk of preterm birth and stillbirth. Finally, we stratified the data by maternal age, origin, education, and deprivation to determine if the association between hospital type and the risk of preterm and stillbirth varied with these characteristics.

We performed the analysis in SAS version 9.4 (RRID:SCR_008567, SAS Institute Inc., Cary, NC). The review board of the University of Montreal Hospital Centre provided an ethics waiver as the data sources used in this study were anonymous and informed consent was not required.

## Inclusivity in global research

Additional information regarding the ethical, cultural, and scientific considerations specific to inclusivity in global research is included in S1 Checklist.

## Results

The study comprised 124,670 births among Anglophones residing in the Montreal metropolitan centre, of which 9,776 (7.8%) were preterm and 559 (0.4%) were stillbirths (Table 1). About 21.7% of Anglophone women delivered at the hospital closest to their residence, while 59.5% delivered at a farther English hospital and 18.0% delivered at a farther French hospital.

**Table 1. Characteristics of Anglophone women at the time of delivery.**

|  | No. births (%) |
|---|---|
| Preterm birth | 9,776 (7.8) |
| Stillbirth | 559 (0.4) |
| Hospital of delivery |  |
| Farther French hospital | 22,401 (18.0) |
| Farther English hospital | 74,215 (59.5) |
| Closest hospital to home | 27,040 (21.7) |
| Maternal age, years |  |
| <25 | 14,600 (11.7) |
| 25–34 | 77,294 (62.0) |
| ≥35 | 32,776 (26.3) |
| Parity |  |
| 0 | 57,052 (45.8) |
| 1 | 42,916 (34.4) |
| ≥2 | 24,702 (19.8) |
| Civil status |  |
| Married | 107,021 (85.8) |
| Single or common-law | 11,346 (9.1) |
| Region of origin |  |
| Canadian born | 84,773 (68.0) |
| Foreign born | 39,897 (32.0) |
| Education |  |
| No high school diploma | 4,869 (3.9) |
| High school diploma or vocational training | 55,464 (44.5) |
| University | 55,573 (44.6) |
| Material deprivation |  |
| High | 18,185 (14.6) |
| Middle-high | 17,542 (14.1) |
| Middle | 19,897 (16.0) |
| Middle-low | 25,746 (20.7) |
| Low | 38,288 (30.7) |
| Total | 124,670 (100.0) |

**Table 2. Association between type of hospital and risk of adverse birth outcome.**

| | Preterm birth | | | Stillbirth | | |
|---|---|---|---|---|---|---|
| | No. preterm (%) | Risk ratio (95% confidence interval) | | No. stillborn (%) | Risk ratio (95% confidence interval) | |
| | | Unadjusted | Adjusted[1] | | Unadjusted | Adjusted[a] |
| Farther French hospital | 1,730 (7.7) | 1.23 (1.16–1.32) | 1.21 (1.14–1.30) | 132 (0.6) | 1.70 (1.30–2.20) | 1.67 (1.28–2.18) |
| Farther English hospital | 6,270 (8.4) | 1.35 (1.28–1.42) | 1.36 (1.29–1.44) | 330 (0.4) | 1.28 (1.02–1.61) | 1.36 (1.08–1.71) |
| Closest hospital to home | 1,691 (6.3) | Reference | Reference | 94 (0.3) | Reference | Reference |

[a]Adjusted for maternal age, parity, civil status, region of origin, education, and material deprivation.

On average, women traveled 6 km to reach the hospital closest to home, 13.5 km to reach a farther English hospital, and 14.8 km to reach a farther French hospital. Most women were between 25 and 34 years of age at the time of delivery (62.0%), although a considerable proportion were 35 years or older (26.3%). 32% of Anglophone women were foreign born, while 68% were Canadian born. Around 4% of women had no high school diploma and 15% had high material deprivation.

Women who delivered at hospitals farther from their residence had an elevated risk of preterm birth and stillbirth (Table 2). Compared with delivery at the hospital closest to the residence, delivery at a farther English hospital was associated with 1.36 times the risk of preterm birth (95% CI 1.29–1.44) and stillbirth (95% CI 1.08–1.71). Delivery at a farther French hospital was associated with 1.21 times the risk of preterm birth (95% CI 1.14–1.30) and 1.67 times the risk of stillbirth (95% CI 1.28–2.18).

Maternal age, education, and material deprivation were all associated with the type of hospital used for delivery (Table 3). Relative to 25 to 34 years, women 35 years or older were 1.08

**Table 3. Association between maternal characteristics and type of hospital used for delivery.**

| | No. births (%) | | | Risk ratio (95% confidence interval)[a] | |
|---|---|---|---|---|---|
| | Closest hospital to home | Farther English hospital | Farther French hospital | Farther English hospital | Farther French hospital |
| Maternal age, years | | | | | |
| <25 | 3,682 (25.2) | 7,137 (48.9) | 3,689 (25.3) | 0.93 (0.91–0.94) | 1.02 (0.99–1.04) |
| 25–34 | 17,233 (22.3) | 45,595 (59.0) | 13,781 (17.8) | Reference | Reference |
| ≥35 | 6,125 (18.7) | 21,483 (65.5) | 4,931 (15.0) | 1.08 (1.07–1.09) | 1.03 (1.01–1.06) |
| Region of origin | | | | | |
| Canadian born | 18,144 (21.4) | 51,117 (60.3) | 14,708 (17.3) | Reference | Reference |
| Foreign born | 8,896 (22.3) | 23,098 (57.9) | 7,693 (19.3) | 0.97 (0.96–0.98) | 0.98 (0.96–1.00) |
| Education | | | | | |
| No high school diploma | 1,162 (23.9) | 2,091 (42.9) | 1,580 (32.5) | 0.96 (0.93–0.98) | 1.21 (1.17–1.25) |
| High school diploma or vocational training | 13,663 (24.6) | 30,385 (54.8) | 10,858 (19.6) | Reference | Reference |
| University | 10,555 (19.0) | 36,447 (65.6) | 8,232 (14.8) | 1.10 (1.09–1.11) | 1.05 (1.03–1.07) |
| Material deprivation | | | | | |
| High | 2,951 (16.2) | 11,113 (61.1) | 4,059 (22.3) | 1.14 (1.12–1.15) | 1.30 (1.26–1.35) |
| Middle-high | 3,962 (22.6) | 9,999 (57.0) | 3,505 (20.0) | 1.03 (1.01–1.04) | 1.06 (1.03–1.10) |
| Middle | 4,674 (23.5) | 11,310 (56.8) | 3,718 (18.7) | Reference | Reference |
| Middle-low | 6,602 (25.6) | 14,650 (56.9) | 4,084 (15.9) | 0.96 (0.95–0.97) | 0.86 (0.83–0.89) |
| Low | 8,095 (21.1) | 24,916 (65.1) | 5,010 (13.1) | 1.02 (1.01–1.03) | 0.86 (0.83–0.89) |

[a]Adjusted for maternal age, parity, civil status, region of origin, education, and material deprivation.

times more likely to deliver at a farther English hospital (95% CI 1.07–1.09), while women less than 25 years were 7% less likely to do so (RR 0.93, 95% CI 0.91–0.94). Compared with having a high school diploma, women with university education were 10% more likely to deliver at a farther English hospital (RR 1.10, 95% CI 1.09–1.11), while women with no high school diploma were 4% less likely to do so (RR 0.96, 95% CI 0.93–0.98). Not having a high school diploma was instead more strongly associated with delivery at a farther French hospital (RR 1.21, 95% CI 1.17–1.25). Women with high deprivation were more likely to deliver at a farther English hospital (RR 1.14, 95% CI 1.12–1.15) or a farther French hospital (RR 1.30, 95% CI 1.26–1.35) compared with moderate material deprivation.

Maternal characteristics were associated with the risk of preterm birth and stillbirth (Table 4). Women 35 years or older had 1.25 times the risk of preterm birth (95% CI 1.20–1.31) and 1.35 times the risk of stillbirth (95% CI 1.11–1.64) relative to 25 to 34 years. Compared with Canadian born women, foreign born women had 1.66 times the risk of stillbirth (95% CI 1.40–1.97). Compared with a high school diploma, not having a diploma was associated with preterm birth (RR 1.26, 95% CI 1.15–1.38). Compared with moderate deprivation, women with high material deprivation had an elevated risk of stillbirth (RR 1.43, 95% CI 1.07–1.91).

In analyses stratified by maternal characteristics, delivery at a farther French hospital was more strongly associated with stillbirth, while delivery at a farther English hospital was not as strongly associated with this outcome (Table 5). Among women with no high school diploma, for example, delivery at a farther French hospital was associated with 2.96 times the risk of stillbirth (95% CI 1.06–8.29), but no risk of preterm birth (95% CI 0.85–1.40), compared with delivery at the closest hospital. In contrast, delivery at a farther English hospital was not associated with stillbirth (95% CI 0.49–4.17), but was associated with 1.41 times the risk of preterm birth (95% CI 1.13–1.76). The inverted pattern of greater risk of stillbirth with delivery at a farther French hospital, but greater risk of preterm birth with delivery at a farther English hospital, was present across all maternal characteristics.

**Table 4. Association between maternal characteristics and risk of adverse birth outcomes.**

| | Preterm birth | | Stillbirth | |
|---|---|---|---|---|
| | No. preterm (%) | Risk ratio (95% confidence interval)[1] | No. stillborn (%) | Risk ratio (95% confidence interval)[a] |
| Maternal age, years | | | | |
| <25 | 1,170 (8.0) | 0.95 (0.89–1.02) | 90 (0.6) | 1.27 (0.99–1.63) |
| 25–34 | 5,661 (7.3) | Reference | 304 (0.4) | Reference |
| ≥35 | 2,945 (9.0) | 1.25 (1.20–1.31) | 165 (0.5) | 1.35 (1.11–1.64) |
| Region of origin | | | | |
| Canadian born | 6,577 (7.8) | Reference | 295 (0.3) | Reference |
| Foreign born | 3,199 (8.0) | 1.00 (0.95–1.03) | 264 (0.7) | 1.66 (1.40–1.97) |
| Education | | | | |
| No high school diploma | 504 (10.4) | 1.26 (1.15–1.38) | 30 (0.6) | 1.28 (0.87–1.88) |
| High school diploma or vocational training | 4,410 (8.0) | Reference | 235 (0.4) | Reference |
| University | 4,022 (7.2) | 0.92 (0.88–0.96) | 176 (0.3) | 0.79 (0.64–0.98) |
| Material deprivation | | | | |
| High | 1,564 (8.6) | 1.04 (0.97–1.12) | 116 (0.6) | 1.43 (1.07–1.91) |
| Middle-high | 1,428 (8.1) | 1.02 (0.95–1.10) | 106 (0.6) | 1.50 (1.12–2.01) |
| Middle | 1,572 (7.9) | Reference | 78 (0.4) | Reference |
| Middle-low | 2,011 (7.8) | 1.00 (0.94–1.07) | 113 (0.4) | 1.14 (0.86–1.53) |
| Low | 2,816 (7.4) | 0.95 (0.90–1.01) | 130 (0.3) | 0.95 (0.72–1.27) |

[a]Adjusted for maternal age, parity, civil status, region of origin, education, and material deprivation.

**Table 5. Association between hospital type and risk of adverse birth outcomes stratified by maternal characteristics.**

| | Risk ratio (95% confidence interval)[a] | | | |
| --- | --- | --- | --- | --- |
| | Farther English hospital | | Farther French hospital | |
| | Preterm birth | Stillbirth | Preterm birth | Stillbirth |
| Maternal age, years | | | | |
| <25 | 1.17 (1.02–1.34) | 2.10 (1.11–3.98) | 1.09 (0.92–1.28) | 2.71 (1.38–5.32) |
| 25–34 | 1.34 (1.26–1.44) | 1.34 (0.98–1.81) | 1.18 (1.08–1.28) | 1.43 (0.99–2.06) |
| ≥35 | 1.52 (1.37–1.68) | 1.13 (0.74–1.73) | 1.40 (1.23–1.60) | 1.68 (1.03–2.74) |
| Region of origin | | | | |
| Canadian born | 1.36 (1.28–1.45) | 1.26 (0.92–1.73) | 1.25 (1.16–1.36) | 1.58 (1.09–2.28) |
| Foreign born | 1.37 (1.25–1.50) | 1.40 (1.00–1.97) | 1.15 (1.03–1.28) | 1.71 (1.16–2.51) |
| Education | | | | |
| No high school diploma | 1.41 (1.13–1.76) | 1.43 (0.49–4.17) | 1.09 (0.85–1.40) | 2.96 (1.06–8.29) |
| High school diploma or vocational training | 1.26 (1.17–1.35) | 1.64 (1.14–2.36) | 1.24 (1.14–1.36) | 2.16 (1.44–3.24) |
| University | 1.49 (1.36–1.63) | 1.10 (0.74–1.66) | 1.28 (1.14–1.44) | 1.49 (0.92–2.41) |
| Material deprivation | | | | |
| High | 1.20 (1.04–1.38) | 1.32 (0.76–2.32) | 1.16 (0.99–1.36) | 1.67 (0.91–3.09) |
| Middle-high | 1.32 (1.16–1.51) | 1.59 (0.94–2.68) | 1.18 (1.01–1.39) | 1.54 (0.83–2.85) |
| Middle | 1.44 (1.27–1.65) | 1.57 (0.83–2.94) | 1.46 (1.25–1.70) | 2.54 (1.28–5.02) |
| Middle-low | 1.42 (1.28–1.59) | 1.29 (0.81–2.08) | 1.21 (1.05–1.40) | 1.39 (0.78–2.50) |
| Low | 1.42 (1.28–1.57) | 1.31 (0.80–2.14) | 1.27 (1.11–1.45) | 2.01 (1.13–3.58) |

[a]Risk ratio for hospital type relative to the closest hospital, adjusted for maternal age, parity, civil status, region of origin, education, and material deprivation.

## Discussion

In this study of the Anglophone minority of metropolitan Montreal, pregnant women who traveled farther for delivery had a greater risk of preterm birth and stillbirth compared with women who delivered closer to home. However, delivery at a farther French hospital was more strongly associated with the risk of stillbirth than preterm birth, whereas delivery at a farther English hospital was associated with both outcomes equally. Associations persisted in analyses stratified by maternal age, region of origin, education, and material deprivation. The association was present despite the potential ability of Montreal Anglophones to communicate in French to some degree, or present to hospital with an English-speaking spouse or family member. As a result, the findings suggest that basic knowledge of French is insufficient to communicate or understand complicated medical terminology. The findings raise the question of whether access to perinatal care in an individual's own mother tongue may help prevent stillbirth. However, future research is needed to determine if pregnancy-related morbidities requiring tertiary care or hospital quality of care also contribute.

Early delivery is a commonly used intervention to prevent antepartum stillbirth, but this strategy may lead to preterm birth if delivery occurs before 37 weeks of gestation [6]. A study of births in the United States and Canada showed that decreases in the rate of stillbirth were typically accompanied by an increase in the rate of preterm birth [15]. While stillbirth rates can be lowered with preterm delivery, infants may nevertheless be at risk of neonatal morbidity and mortality, especially infants born at very low gestational ages [6]. The benefits of early delivery therefore have to be weighed carefully [6].

We found that Anglophone women had a higher risk of stillbirth when they delivered at a farther French hospital, but a higher risk of preterm birth when they delivered at a farther English hospital. The trend was present in each stratum of maternal age, region of origin,

education group, and level of material deprivation. Patient care may vary depending on the quality of communication with providers [5]. Communication barriers among Anglophones in French hospitals have the potential to lead to delays in labor induction or use of cesarean section to prevent antepartum or intrapartum stillbirth. Communication barriers have been shown to affect access to healthcare services and information, including a patient's understanding of medical terminology [5]. In the United States, where the population of Spanish-speakers increased rapidly in the last decades, health literacy has been identified as a major determinant of health inequality [16]. Language barriers can affect the quality of health services, with research demonstrating an association with delays in surgery [16]. While ability to communicate may explain some of the greater risk of stillbirth in French hospitals, it is also possible that standards of care differ between hospitals [17]. Further research would be needed to determine if English hospitals tend to induce labor earlier than French hospitals. This practice would lead to more preterm births in English hospitals.

Risks of preterm birth and stillbirth were greater for all women who delivered farther from home, regardless of hospital characteristics. The excess risk is likely due to the presence of comorbidities requiring tertiary care at farther hospitals, although travel time may be a contributing factor. Distance and time travel have been associated with a greater risk of adverse pregnancy outcomes in previous research [10–12, 18, 19]. A retrospective cohort study of 820,000 Canadian women found that travel distances of 50 km or more to reach a birth hospital were associated with maternal morbidity, stillbirth, and early neonatal mortality, compared with travel distances of 20 to 49 km [18]. A study of births from France found that stillbirth rates were higher for women who traveled ≥45 km from home [19]. A Dutch study of 1 million singletons born at term found that women who traveled at least 20 minutes from home had 1.5 times greater odds of neonatal mortality [12]. In Wales, every 15 minute increase in travel time was associated with 1.15 times the likelihood of intrapartum and neonatal mortality [10]. In our data, delivery at a farther hospital was associated with up to 1.4 times the risk of preterm birth and 1.7 times the risk of stillbirth.

Distance may influence the choice of birth hospital [17, 20]. Data suggest that women generally consider shorter distances safer as they can obtain hospital care more quickly [17]. Costs of transportation can also influence the choice of hospital [5]. Nevertheless, qualitative studies indicate that women may be willing to travel longer distances to access higher quality perinatal care [17, 20]. In a Danish study, women with high-risk pregnancies tended to prefer university hospitals over regional hospitals because of the higher level of specialization, even if it meant having to travel farther [17]. In our study, Anglophone women with fewer morbidities and less need for specialized care may have opted to deliver closer to home. In contrast, women who traveled farther from home may have required more high-risk perinatal care.

Disadvantaged women may benefit the most from perinatal services in their mother tongue, as socioeconomic deprivation is one of the strongest risk factors for preterm birth and stillbirth [21–23]. Previous studies have shown that socioeconomically disadvantaged Anglophones have particularly high rates of stillbirth and preterm birth in Quebec [1, 2]. In our data, Anglophone women with no high school diploma or with a high level of material deprivation were more likely to deliver at a farther French hospital than Anglophone women who were not disadvantaged. These findings are concerning as communication barriers may be more prevalent in disadvantaged women. In this study, Anglophones with no high school diploma were most at risk of preterm birth, and women with high or moderately high deprivation had up to 1.5 times the risk of stillbirth. Better access to perinatal care in woman's language could potentially prevent adverse pregnancy outcomes in disadvantaged populations.

This study had limitations. We used an administrative registry and cannot rule out coding errors that may have resulted in nondifferential misclassification of exposures or outcomes,

and attenuated associations toward the null. We identified Anglophones using self-reported mother tongue, which may be associated with some degree of misclassification. We could not determine whether women were comfortable understanding or conversing in French. We lacked data on residual confounders such as individual income, duration of residence in Canada, and planned place of birth. We could not adjust for multiple birth or the added variance due to clustering of births within mothers. We could not consider level of care of hospitals and maternal medical characteristics which may have influenced the choice of hospital. The Anglophone population may comprise cultural groups with different preferences or risks regarding perinatal care, although we accounted for migrant status. The results pertain to the Anglophone minority of Montreal, and may not be generalizable to minority populations in different settings.

## Conclusions

This study of the Anglophone minority in metropolitan Montreal found that women who traveled to a farther French hospital had a greater risk of stillbirth than preterm birth, whereas women who traveled to a farther English hospital had no meaningful difference in these outcomes. Nevertheless, all women who delivered farther from their residence had a higher risk of preterm birth and stillbirth than women who delivered closer to home. Associations were present regardless of maternal age, education, and socioeconomic status. The findings highlight the importance of communication in perinatal healthcare for minority language groups. Having access to a hospital that provides care in the main language of choice has potential to reduce the risk of stillbirth.

## Supporting information

**S1 Checklist. Questionnaire on inclusivity in global research.**
(PDF)

## Author Contributions

**Conceptualization:** Nathalie Auger, Marianne Bilodeau-Bertrand, Nahantara Lafleur.

**Formal analysis:** Marianne Bilodeau-Bertrand.

**Funding acquisition:** Nathalie Auger.

**Methodology:** Nathalie Auger, Marianne Bilodeau-Bertrand.

**Supervision:** Nathalie Auger.

**Visualization:** Nahantara Lafleur.

**Writing – original draft:** Marianne Bilodeau-Bertrand.

**Writing – review & editing:** Nathalie Auger, Marianne Bilodeau-Bertrand, Nahantara Lafleur.

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
