## [Decision Letter · Decision Letter 0]

6 Mar 2023

PONE-D-22-31177Access to perinatal healthcare in minority Anglophones: Hospital type and birth outcomePLOS ONE

Dear Dr. Auger,

Thank you for submitting your manuscript to PLOS ONE. After careful consideration, we feel that it has merit but does not fully meet PLOS ONE’s publication criteria as it currently stands. Therefore, we invite you to submit a revised version of the manuscript that addresses the points raised during the review process.

ACADEMIC EDITOR:Kindly refer thoroughly to the reviewers' comments, especially reviewer's 2 comments on statistical issues. We ask you to give a special attention to the methodology including the statistical analysis, the data provided as basis for the findings, and the results' presentation. 

We look forward to receiving your revised manuscript.

Kind regards,

Suhad Daher-Nashif, MSc., PhD

Academic Editor

PLOS ONE

Journal Requirements:

3. Please include a complete copy of PLOS’ questionnaire on inclusivity in global research in your revised manuscript. Our policy for research in this area aims to improve transparency in the reporting of research performed outside of researchers’ own country or community. The policy applies to researchers who have travelled to a different country to conduct research, research with Indigenous populations or their lands, and research on cultural artefacts. The questionnaire can also be requested at the journal’s discretion for any other submissions, even if these conditions are not met.  Please find more information on the policy and a link to download a blank copy of the questionnaire here: https://journals.plos.org/plosone/s/best-practices-in-research-reporting. Please upload a completed version of your questionnaire as Supporting Information when you resubmit your manuscript.

Reviewers' comments:

Reviewer's Responses to Questions

**Comments to the Author**

1. Is the manuscript technically sound, and do the data support the conclusions?

Reviewer #1: Yes

Reviewer #2: No

2. Has the statistical analysis been performed appropriately and rigorously? 

Reviewer #1: Yes

Reviewer #2: No

3. Have the authors made all data underlying the findings in their manuscript fully available?

Reviewer #1: Yes

Reviewer #2: No

4. Is the manuscript presented in an intelligible fashion and written in standard English?

Reviewer #1: Yes

Reviewer #2: Yes

5. Review Comments to the Author

Reviewer #1: The authors presented some interesting data on whether the hospital’s primary language of service (French or English) may affect birth outcomes for mothers whose native language is English (Anglophone) – a large minor language group in Montreal. They found that minority Anglophones in Montreal who travel to a farther French hospital for delivery have a greater risk of stillbirth than Anglophones who travel to a farther English hospital. Distance to the place of delivery affect birth outcomes. Delivery at a farther hospital (whether English or French) was associated with higher risks of preterm birth and stillbirth than delivery at a hospital close to home. The study is based on a large birth cohort including 124,755 births among Anglophones in metropolitan Montreal.

In general, the study was well conducted, and the manuscript is well-written.

I only have some minor edits:

1. Title, change “birth outcome” to “birth outcomes”

2. Abstract, line 23, change “risk of ” to “risks of”

3. Abstract, last sentence, change “More effort is needed” to “This novel observation suggests the need to”

4. Keyword: change “Premature birth” to “preterm birth”

5. Introduction, line 43, change “one systematic review” to “a systematic review”

6. Page 8, Table 1, for civil status, I believe “married” should be “married or common-law union”, and for education, delete “training” after “University”

7. Discussion, page 14, line 241, change “Risk of preterm birth and stillbirth was greater …” to “Risks of preterm birth and stillbirth were greater …”

Reviewer #2: This study examined and compared differences in PTB and stillbirth occurrences between Anglophone women who delivered at a farther English speaking or French speaking hospital (than the closest available from their home) and those who delivered at the closest hospital in Montreal, Canada. The study was based on 124,755 births in 1998 – 2019 identified from live birth and still registration databases to identify maternal mother tongue (Anglophone or not), postal code of residence, hospital of birth, and the study outcomes, and found the risks of PTB and stillbirth were slightly higher among women who delivered at a farther hospital. The study needs to address some important points to clarify its importance and contributions.

The exposure of interest is not clearly defined in regard to what it represents. What is the exposure of interest exactly? The authors discussed several points related to traveling to a hospital for delivery in remote distance in Discussion, but the study exposure is not about how far but the primary language of service at a farther hospital. Communication barriers is one of things that the authors hint what it might capture, but the exposure variable used in the study does not necessarily capture that. The mother tongue (English) recorded on the birth certificate does not necessarily mean that the mother is not capable of communicating in French. Even if it is the case, their spouse or family member accompanying might be able to communicate in French, and the doctors/staffs can communicate in English at a “French” hospital (and often it is the case in Montreal).

In addition, it is unclear how communication language during labour and delivery would affect preterm birth and stillbirth? The authors claim that “[C]ommunication barriers among Anglophones in French hospitals have the potential to lead to delays in labor induction or use of cesarean section to prevent stillbirth.” in Discussion. (p.13) However, this requires the stillbirth outcome of the study mainly to be the death during labour, which is not the case according to its definition in the study (see my comment below). There is also no support in the study to conceptualize the delivery hospital was the hospital where the mothers had prenatal care all along.

The stillbirth is defined based solely on the birth weight criterion (<=500g) that is the provincial criterion. However, it is well known that it includes both viable and non-viable births, and late pregnancy termination, thus a combination of true fetal deaths and elected late abortion. This is also the case for a stillbirth definition by gestational age that is commonly used. The authors mainly discuss the outcome of stillbirth as a proxy of death during labour/delivery, but the actual measure used is far from that. Please clarify what is the stillbirth outcome that is hypothesized to be affected by the language and distance of a delivery hospital.

The main conclusion is quite misleading: “This study of the Anglophone minority in metropolitan Montreal found that women who traveled to a farther French hospital had a greater risk of stillbirth, while women who traveled to a farther English hospital had a greater risk of preterm birth.” This conclusion is mainly drawn from the difference in point estimates it seems. The Cis overlap with each other and the authors also stated that the differences across outcomes were not statistically significant (p.13). Even if they were, the differences are so minimal to have meaningful impacts—They did not present differences on absolute scale, but the stillbirth was very rare in occurrence (Table 1). In addition, as described above, there are insufficient explanations and data to support their explanations/arguments.

Please clarify whether the analysis was restricted to singleton births where both PTB and stillbirth risks are higher. Also, there would be multiple births to the same mother over the study period, 1998 – 2019. Please show the extent of clustering at mother and how it was accounted for.

6. PLOS authors have the option to publish the peer review history of their article (what does this mean?). If published, this will include your full peer review and any attached files.

Reviewer #1: No

Reviewer #2: No

---

## [Author Response · Author response to Decision Letter 0]

15 Mar 2023

RESPONSE TO REVIEW

ACADEMIC EDITOR:

E1.1. Kindly refer thoroughly to the reviewers' comments, especially reviewer's 2 comments on statistical issues. We ask you to give a special attention to the methodology including the statistical analysis, the data provided as basis for the findings, and the results' presentation.

Response: We thank the Editor. We clarified the statistical analysis, basis of the data, and presentation of results throughout, and did our best to address other points raised by the Reviewers.

Reviewer #1:

The authors presented some interesting data on whether the hospital’s primary language of service (French or English) may affect birth outcomes for mothers whose native language is English (Anglophone) – a large minor language group in Montreal. They found that minority Anglophones in Montreal who travel to a farther French hospital for delivery have a greater risk of stillbirth than Anglophones who travel to a farther English hospital. Distance to the place of delivery affect birth outcomes. Delivery at a farther hospital (whether English or French) was associated with higher risks of preterm birth and stillbirth than delivery at a hospital close to home. The study is based on a large birth cohort including 124,755 births among Anglophones in metropolitan Montreal.

R1.1. In general, the study was well conducted, and the manuscript is well-written.

Response: We thank the Reviewer.

R1.2. Title, change “birth outcome” to “birth outcomes”.

Response: We modified the title (line 1).

R1.3. Abstract, line 23, change “risk of ” to “risks of”.

Response: We made the requested change (line 23).

R1.4. Abstract, last sentence, change “More effort is needed” to “This novel observation suggests the need to”.

Response: We corrected the sentence (line 35).

R1.5. Keyword: change “Premature birth” to “preterm birth”

Response: We replaced the keyword (line 38).

R1.6. Introduction, line 43, change “one systematic review” to “a systematic review”.

Response: We made the correction (line 43).

R1.7. Page 8, Table 1, for civil status, I believe “married” should be “married or common-law union”, and for education, delete “training” after “University”.

Response: As the data were based on the formal definition of civil status, we referred to legally married women only. We clarified that we included “legal civil status (married, single or common-law)” (line 116). We also deleted the term training from Table 1.

R1.8. Discussion, page 14, line 241, change “Risk of preterm birth and stillbirth was greater …” to “Risks of preterm birth and stillbirth were greater …”

Response: We modified the sentence and thank the Reviewer (line 254).

Reviewer #2:

R2.1. This study examined and compared differences in PTB and stillbirth occurrences between Anglophone women who delivered at a farther English speaking or French speaking hospital (than the closest available from their home) and those who delivered at the closest hospital in Montreal, Canada. The study was based on 124,755 births in 1998 – 2019 identified from live birth and still registration databases to identify maternal mother tongue (Anglophone or not), postal code of residence, hospital of birth, and the study outcomes, and found the risks of PTB and stillbirth were slightly higher among women who delivered at a farther hospital. The study needs to address some important points to clarify its importance and contributions.

Response: We thank the Reviewer and clarified that “The findings highlight the importance of communication in perinatal healthcare for minority language groups. Having access to a hospital that provides care in the main language of choice has potential to reduce the risk of stillbirth” (lines 310-313).

R2.2. The exposure of interest is not clearly defined in regard to what it represents. What is the exposure of interest exactly? The authors discussed several points related to traveling to a hospital for delivery in remote distance in Discussion, but the study exposure is not about how far but the primary language of service at a farther hospital. Communication barriers is one of things that the authors hint what it might capture, but the exposure variable used in the study does not necessarily capture that. The mother tongue (English) recorded on the birth certificate does not necessarily mean that the mother is not capable of communicating in French. Even if it is the case, their spouse or family member accompanying might be able to communicate in French, and the doctors/staffs can communicate in English at a “French” hospital (and often it is the case in Montreal).

Response: The exposure of interest is the language of the hospital. English mother tongue is an inclusion criterion rather than an exposure. We clarified that “The main exposure measure was the primary language of the hospital used for delivery. Women could deliver at the hospital closest to their residence or at a farther hospital that offered either English or French perinatal services. Although French hospitals can provide care in English, staff may not be as fluent or find it as simple to identify urgent care needs. Ease of communication may not be as fluid for Anglophone women requiring tertiary care at a French hospital” (lines 77-81). We also added that “The association was present despite the potential ability of Montreal Anglophones to communicate in French to some degree, or present to hospital with an English-speaking spouse or family member. As a result, the findings suggest that basic knowledge of French is insufficient to communicate or understand complicated medical terminology” (lines 220-224). The ease with which a woman may converse in French varies from person to person, whether or not a spouse or family member is present. Some Anglophones are comfortable using French, while others are not. Ability to understand and speak French does not mean that there is full comprehension of the language. We added the limitation that “We could not determine whether women were comfortable understanding or conversing in French” (lines 294-295).

R2.3. In addition, it is unclear how communication language during labour and delivery would affect preterm birth and stillbirth? The authors claim that “[C]ommunication barriers among Anglophones in French hospitals have the potential to lead to delays in labor induction or use of cesarean section to prevent stillbirth.” in Discussion. (p.13) However, this requires the stillbirth outcome of the study mainly to be the death during labour, which is not the case according to its definition in the study (see my comment below). There is also no support in the study to conceptualize the delivery hospital was the hospital where the mothers had prenatal care all along.

Response: The Reviewer is correct that some fetal deaths may occur before the onset of labor. However, stillbirth during labor is not the outcome of interest. Prevention of stillbirth begins before the onset of labor, as women may experience antepartum signs and symptoms that predict fetal death. Pregnant women are instructed to call the hospital for any problem or emergency. Women who incorrectly communicate their problem or incorrectly interpret medical advice are at risk if there is a delay in labor induction. We clarified that “Stillbirths include fetal deaths that occur antepartum or intrapartum. Antepartum fetal deaths can potentially be prevented with labor induction, whereas intrapartum deaths can be prevented with cesarean delivery” (lines 104-106). The delivery hospital is decided by the physician prenatally. The physician follows the patient in the prenatal period, and sends the contents of the obstetric chart to the hospital before delivery. The patient delivers at the designated hospital where the chart is located, rather than at another hospital.

R2.4. The stillbirth is defined based solely on the birth weight criterion (<=500g) that is the provincial criterion. However, it is well known that it includes both viable and non-viable births, and late pregnancy termination, thus a combination of true fetal deaths and elected late abortion. This is also the case for a stillbirth definition by gestational age that is commonly used. The authors mainly discuss the outcome of stillbirth as a proxy of death during labour/delivery, but the actual measure used is far from that. Please clarify what is the stillbirth outcome that is hypothesized to be affected by the language and distance of a delivery hospital.

Response: We agree with the Reviewer and have removed late pregnancy terminations from the analysis. We updated all the results in the manuscript and have clarified that stillbirth was defined “as the intrauterine death of a fetus weighing 500g or more that was not due to voluntary termination” (lines 108-109). This change did not alter the interpretation of results.

We addressed viable and non-viable stillbirths as follows: “Stillbirths include fetal deaths that occur antepartum or intrapartum. Antepartum fetal deaths can potentially be prevented with labor induction, whereas intrapartum deaths can be prevented with cesarean delivery” (lines 104-106). In the discussion, we clarified that labor induction and cesarean section can be used to prevent “antepartum or intrapartum stillbirth” (line 243).

R2.5. The main conclusion is quite misleading: “This study of the Anglophone minority in metropolitan Montreal found that women who traveled to a farther French hospital had a greater risk of stillbirth, while women who traveled to a farther English hospital had a greater risk of preterm birth.” This conclusion is mainly drawn from the difference in point estimates it seems. The Cis overlap with each other and the authors also stated that the differences across outcomes were not statistically significant (p.13). Even if they were, the differences are so minimal to have meaningful impacts—They did not present differences on absolute scale, but the stillbirth was very rare in occurrence (Table 1). In addition, as described above, there are insufficient explanations and data to support their explanations/arguments.

Response: The conclusion is drawn from the associations for women who deliver at a farther French hospital, where the risk of stillbirth is higher than the risk of preterm birth. We restructured Table 5 to make the comparison of these outcomes clearer. Table 5 shows that among French hospitals, Anglophones in every stratum always have a higher risk of stillbirth than preterm birth. This pattern is absent in English hospitals, where the risk of stillbirth is either the same as preterm birth, or sometimes higher or lower. Focusing on statistical significance is not recommended in the literature (Greenland 2016; Wasserstein 2019). For this reason, we do not feel justified suggesting that women who deliver at a farther French hospital have similar risks as those who deliver at a farther English hospital. Instead, we modified the conclusion as follows: “This study of the Anglophone minority in metropolitan Montreal found that women who traveled to a farther French hospital had a greater risk of stillbirth than preterm birth, whereas women who traveled to a farther English hospital had no meaningful difference in these outcomes” (lines 305-307).

Greenland S, Senn SJ, Rothman KJ, et al. Statistical tests, P values, confidence intervals, and power: a guide to misinterpretations. Eur J Epidemiol. 2016;31(4):337–50.

Wasserstein RL, Schirm AL, Lazar NA. Moving to a world beyond “p < 0.05.” Am Stat. 2019;73(sup1):1–19.

R2.6. Please clarify whether the analysis was restricted to singleton births where both PTB and stillbirth risks are higher. Also, there would be multiple births to the same mother over the study period, 1998 – 2019. Please show the extent of clustering at mother and how it was accounted for.

Response: The analysis included all births, whether singleton or higher order. Unfortunately, attempts to adjust for multiple birth did not allow models to converge, likely because multiple birth was rare (3.6% of births were multiple). As we did not have a unique patient identifier, we also could not adjust for clustering within mothers. We added the limitation that “We could not adjust for multiple birth or the added variance due to clustering of births within mothers” (lines 297-298). We thank the Reviewer.

---

## [Editor Report · Decision Letter 1]

4 Apr 2023

Access to perinatal healthcare in minority Anglophones: Hospital type and birth outcomes

PONE-D-22-31177R1

Dear Dr. Auger,

We’re pleased to inform you that your manuscript has been judged scientifically suitable for publication and will be formally accepted for publication once it meets all outstanding technical requirements.

Kind regards,

Suhad Daher-Nashif, MSc., PhD

Academic Editor

PLOS ONE

---

## [Editor Report · Acceptance letter]

10 Apr 2023

PONE-D-22-31177R1 

Access to perinatal healthcare in minority Anglophones: Hospital type and birth outcomes 

Dear Dr. Auger:

I'm pleased to inform you that your manuscript has been deemed suitable for publication in PLOS ONE. Congratulations! Your manuscript is now with our production department. 

Kind regards, 

on behalf of

Dr. Suhad Daher-Nashif 

Academic Editor

PLOS ONE